# Persistent Symptoms in Post-COVID-19 Patients Attending Follow-Up OPD at Sukraraj Tropical and Infectious Disease Hospital (STIDH), Kathmandu, Nepal

**DOI:** 10.3390/tropicalmed6030113

**Published:** 2021-06-28

**Authors:** Anup Bastola, Richa Nepal, Bikesh Shrestha, Kijan Maharjan, Sanjay Shrestha, Bimal Sharma Chalise, Jenish Neupane

**Affiliations:** 1Department of Tropical Medicine, Sukraraj Tropical and Infectious Disease Hospital, Teku, Kathmandu 44600, Nepal; docanup11@gmail.com; 2Department of Internal Medicine, Sukraraj Tropical and Infectious Disease Hospital, Teku, Kathmandu 44600, Nepal; Bkeshshrestha4@gmail.com (B.S.); Kijan1069@gmail.com (K.M.); Shrestha834@gmail.com (S.S.); bschalise@gmail.com (B.S.C.); drneupanejenish@gmail.com (J.N.)

**Keywords:** COVID-19, Nepal, post-acute COVID-19 syndrome

## Abstract

The long-term effects of COVID-19 among survivors is a matter of concern. This research aimed to study persistent symptoms in post-COVID-19 patients attending a follow-up clinic at a tertiary care hospital in Nepal. All patients, presenting to the outpatient clinic during the study duration of six weeks, with history of positive reverse transcriptase- polymerase chain reaction for severe acute respiratory syndrome-coronavirus-2 (SARS-CoV-2) at least two weeks prior to presentation, were included. The duration of follow-up ranged from 15 till 150 days with the mean duration of 28 days after diagnosis of COVID-19. Of 118 patients, 43 (36.4%) had a history of mild COVID-19, 15 (12.8%) had moderate, and 60 (50.8%) had severe. At the time of presentation, 97 (82.2%) patients reported that they had at least one persistent/new symptom beyond two weeks from the diagnosis of COVID-19. Dyspnea, fatigue, chest heaviness, and cough were the commonest persistent complaints in 48 (40.7%), 39 (33.1%), 33 (28%), and 32 (27.1%) patients, respectively. The findings in our study highlight the need for extended monitoring of post-COVID-19 patients following discharge, in order to understand and mitigate long-term implications of the disease.

## 1. Introduction

More than a year after reporting Coronavirus Disease-2019 (COVID-19) for the first time in China, the disease is unrelenting, with more than 147 million cases and 3.1 million deaths worldwide, as of 28 April 2021 [1]. Persistent/new symptoms following convalescence of the clinical disease and/or microbiological recovery have been observed in a large fraction of COVID-19 patients [2]. To date, there is no clear consensus on the definition of such a group of patients. Long COVID, long haul COVID, chronic COVID syndrome, post-COVID-19 syndrome, and post-acute COVID-19 are the different terminologies currently in use for the condition characterized by the persistence of symptoms beyond the acute phase of COVID-19 [3].

Post-COVID-19 patients continue to have persistent symptoms or may have new symptoms following apparent clinical recovery as a result of the consequences of organ damage during acute COVID-19 infection, neurobehavioral abnormalities due to the disease process or hospital admission and intensive care strategies. A perplexing finding noted by physicians in several parts of the world was the prevalence of long COVID unrelated to severity of illness of COVID-19, unlike other diseases, where rigorous intensive care strategies in severely ill patients were associated with long-term health implications in survivors [4].

There have been unprecedented efforts from the scientific community worldwide for the diagnosis, treatment, and prevention of COVID-19; however, little has been done to address the potential long-term health implications for more than 95% COVID-19 patients who have recovered from this novel disease worldwide. Signs and symptoms that fail to return to a healthy baseline status beyond two weeks from the disease onset could be considered to a long-term effect of COVID-19 in apparently recovered post-COVID-19 patients [5]. This research aims to study persistent or new symptoms in post-COVID-19 patients presenting to a tertiary level infectious disease hospital in central Nepal.

## 2. Materials and Methods

### 2.1. Study Design and Settings

This was a descriptive, cross-sectional study conducted among post-COVID-19 patients who had presented to the follow-up outpatient department of a central level infectious disease government hospital, Sukraraj Tropical and Infectious Disease Hospital (STIDH) in Kathmandu, the capital city of Nepal. All patients diagnosed with COVID-19 by positive real time polymerase chain reaction (RT-PCR) for SARS-CoV-2, at least two weeks prior to presentation, were included in our study. Data collection was done over the period of six weeks from 1 March 2021 to 14 April 2021. All patients who had been in home isolation or were discharged following hospital admission from STIDH or from any other hospital during their acute COVID-19 illness were included in this study.

### 2.2. Data Collection

Data collection was done through a face-to-face interview by using a standardized structured questionnaire after consent for voluntary participation was obtained. History taking and relevant clinical examination were done by two experienced attending physicians. Patients were asked to answer Yes/No questions for the symptoms related to post-COVID pathology. Data related to course of COVID-19 illness and index admission were collected retrospectively from the patients, and by reviewing past medical records/discharge sheets brought by patients in the follow-up OPD. Cronbach’s alpha was computed for reliability analysis of the questionnaire using responses from the first fifty consecutive participants. The value was calculated as 0.84 and found to be acceptable.

Patients were asked to report newly occurring or persistent symptoms, or any other symptom worse than before COVID-19 development. The symptoms that did not return to baseline and lasted for more than two weeks following diagnosis of COVID-19 were labeled as persistent symptoms and those symptoms that appeared after two weeks of diagnosis of COVID-19, and were not attributed to other diseases, were labeled as new symptoms in post-COVID-19 patients. Dyspnea/shortness of breath, fatigue, chest heaviness, cough, chest pain, palpitations, anosmia/hyposmia, ageusia/hypogeusia, decreased appetite, headache, and throat discomfort were the persistent symptoms reported by post-COVID-19 patients in our study. Likewise, insomnia, anxiety, hot flushes, parosmia, burning sensation along limbs, impaired concentration, and burning sensation in the perinostrillar area were the new symptoms reported by post-COVID-19 patients in our study. Level of dyspnea/shortness of breath were defined in terms of the Modified Medical Research Council (MMRC) dyspnea scale, where higher scores corresponded to increased level of dyspnea.

### 2.3. Data Analysis

Data was entered in Microsoft Excel and then analyzed with a statistical package for social sciences (SPSS) version 25. Descriptive analyses were done using frequencies and percentages for categorical variables, and means and standard deviation for continuous variables. Bivariate analysis was done using Chi square and Fisher’s exact test, where appropriate. *p* value < 0.05 was considered to be significant.

## 3. Results

Out of 118 patients enrolled in our study, 81 (68.6%) were males and 37 (31.4%) were females with the male: female ratio of approximately 2.2:1. The mean age of presentation was 49.7 ± 15.01 years with minimum age of 25 years and maximum age of 87 years. Thirty-four (28.8%) patients were aged 60 and above, whereas the majority of patients in our study were in the age group 30 to 59 years (64.4%). The most common comorbidity among post-COVID-19 patients who had presented to the follow-up OPD in our study was hypertension (24.6%), followed by diabetes mellitus (14.4%), and chronic respiratory diseases (7.6%). Sixty-three (53.4%) patients did not have any comorbidities in our study. The duration of follow-up of post-COVID-19 patients ranged from 15 to 150 days with mean duration of 28 days (S.D—17.5 days) after diagnosis. More than two-third of the total patients (66.9%) had their first follow-up within three to four weeks of diagnosis of COVID-19, 25 (21.2%) patients presented within five to six weeks, nine (7.6%) patients presented within seven to eight weeks, three (2.5%) patients presented within nine to twelve weeks, and two (1.7%) patients presented after three months of diagnosis of COVID-19 (Table 1).

Table 2 enlists the symptoms during the acute phase of COVID-19 illness, as reported by the study participants in the follow-up OPD of STIDH. The most common symptoms were fever (74.6%), myalgia (74.6%), cough (73.7%), and dyspnea (60.2%). Anosmia/hyposmia and ageusia/dysgeusia were reported by 49.2% and 45.8% patients, respectively. forty-three (36.4%) patients had mild COVID-19, 15 (12.8%) had moderate COVID-19, and 60 (50.8%) had severe COVID-19 in our study. Out of the total number of patients, 34 (28.8%) were admitted to the ward and 38 (32.2%) to intensive care units during the acute phase of COVID-19 infection (Table 3). The mean duration of hospital admission for patients who got admitted was 9.43 ± 5.76 days with minimum duration of one day and maximum duration of 33 days.

At the time of presentation of the first follow-up at STIDH OPD, 97 (82.2%) patients had reported that they had at least one persistent/new symptom beyond two weeks from diagnosis of COVID-19 infection (Figure 1). Twenty-three (19.5%) patients had one symptom, 52 (44.1%) had two to four symptoms, and 22 (18.6%) had five or more symptoms at their first follow-up visit. Twenty-one (17.8%) patients did not have any complaint during their follow-up visit at STIDH (Figure 2). Table 4 enlists the symptoms that were persistent/had newly emerged following acute disease in the post-COVID-19 patients in our study. Dyspnea, fatigue, chest heaviness, and cough were the commonest complaints present in 48 (40.7%), 39 (33.1%), 33 (28%), and 32 (27.1%) patients, respectively. Insomnia and anxiety was present in 19 (16.1%) and 16 (13.6%) post-COVID patients. Anosmia/hyposmia was persistent in 11 (9.3%) and ageusia/hypogeusia was persistent in 8 (6.8%) patients during their first follow-up visit (Table 4).

Out of 48 patients in our study who reported dyspnea to be persistent at their first follow-up visit, 29 (24.6%) had MMRC 1 symptoms, 14 (11.9%) had MMRC 2 symptoms, 5 (4.2%) had MMRC 3 symptoms, and none had MMRC 4 symptoms. Resting hypoxia (room air oxygen saturation less than 94%) was recorded in 24 (20.33%) patients, out of which 15 (12.7%) had mild hypoxia (oxygen saturation 90 to 93%), 3 (2.5%) had moderate hypoxia (oxygen saturation 85 to 89%), and 6 (5.1%) had severe hypoxia (oxygen saturation less than 85%). Sixteen (13.6%) patients who had resting hypoxia were using domiciliary oxygen therapy at home, though intermittently (Table 5).

On bivariate analysis of the presence of at least one persistent/new symptom in post-COVID-19 patients to some pertinent baseline characteristics, no significant association was demonstrated with age groups, gender, presence of any comorbidity, or severity of acute COVID-19 illness (*p* value- 0.177, 0.463, 0.919, and 0.056, respectively). However, the presence of at least one persistent/new symptom in post-COVID-19 patients was significantly associated to mode of isolation during their acute COVID-19 illness (*p* value 0.040) (Table 6).

## 4. Discussion

As the COVID-19 pandemic continues unabated, a vast majority of survivors have presented to healthcare providers with a multitude of signs and symptoms representing possible long-term effects following acute COVID-19 infection [6]. With the persistence of debilitating complaints by the survivors of previous coronavirus infections like Severe Acute Respiratory Syndrome (SARS) in 2003 and Middle East Respiratory Syndrome (MERS) in 2012, the current concern regarding long haulers of COVID-19 seems justified [7]. In a systematic review on the long-term effects of COVID-19 by Leon et al., 80% post-COVID-19 patients (95% confidence interval 65–92%) continue to have one or more symptoms following two weeks of acute COVID-19 infection [5]. Another study by Morin et al. reported that 51% patients had at least one symptom after four months of diagnosis of COVID-19 in a cohort of 478 hospitalized patients [8]. Similarly, in a review by Pavli et al., the incidence of post-COVID syndrome was estimated to be 10 to 35% among patients who were treated on an outpatient basis, and reached 85% among patients who were hospitalized [9]. In our study, around 82.2% patients, with a recent history of confirmed COVID-19 infection, continue to have at least one persistent/new symptom beyond two weeks of diagnosis.

Shortness of breath was the commonest complaint reported by 40.7% patients at mean follow-up duration of 28 days in our study. In a study done in Italy by Carfi et al., fatigue (53.1%) and dyspnea (43.4%) were the commonest complaints of post-COVID-19 patients at mean follow-up duration of 60 days [10]. Persistent dyspnea could result due to the underlying pathology of impaired diffusion capacity, impaired respiratory muscle strength, and fibrotic abnormalities in post-COVID lungs, which were more frequently encountered in severe forms of COVID-19 following convalescence [11]. Another study from China reported fatigue (63%) and sleep disturbances (26%) to be persistent among post-COVID-19 patients at the median duration of six months from the disease onset [12]. Our study reported fatigue in 33.1% patients, whereas insomnia and anxiety were persistent in 16.1% and 13.6% patients during their first follow-up visit following COVID-19. Post viral fatigue has been postulated to result from immune dysregulation and autonomic alterations following COVID-19 infection [13]. Psychological effects following recovery from acute COVID-19 illness could result due to direct viral effects on cognition or may result from the social circumstances related to disease like loss of loved ones, spanning fears about future, job loss, anticipation anxiety, lockdown, and intensive care strategies following which the patient survived [14].

Neurological symptoms like anosmia/hyposmia and ageusia/dysgeusia were found to be persistent in 9.3% and 6.8% patients at their first follow-up visit. One patient reported perception of foul smell and altered smell pattern suggestive of parosmia even after three months of mild COVID-19 infection. Post-infectious olfactory dysfunction in form of misperception of existing odors has been previously reported as a delayed complication following COVID-19 [15]. Similarly, another patient reported a new onset of burning sensation and lancating pain at the left perinostrillar and maxillary area eight weeks following recovery from mild COVID-19 infection, suggestive of left-sided trigeminal neuralgia. The existence of trigeminal neuralgia in relation to SARS-CoV-2 has been recently described in the literature as one of the uncommon manifestations of COVID-19 [16]. Two (1.7%) patients complained of impaired concentration following COVID-19, of which one was below 30 years of age who did not require hospital admission and the other was above 75 years of age, treated in an intensive care unit. ‘Brain fog’ is a colloquial term being used to describe the cognitive difficulties faced by a large group of patients following COVID-19. Neuroinflammation, resulting from the pathogenic and stress stimuli, is the proposed mechanism for this condition, though concrete evidence is yet to be seen [17].

Around two-third (65.3%) of total patients said that they did not feel as normal as before the diagnosis at mean duration of 28 days following COVID-19 infection. Eighteen percent of the total patients reported more than four persistent symptoms during their follow-up visit at STIDH. An important finding in our study was that the presence of at least one persistent/new symptom in post-COVID-19 patients did not correlate with age groups, gender, comorbidities, or severity of COVID-19. Similar findings were also mentioned by Rio et al., who stated that long COVID symptoms did not correlate with chronic comorbidities or severity of acute COVID-19 illness. [18]. Likewise, Terfonde et al. stressed on the persistence of symptoms in younger group of patients without any comorbid conditions, such that could result in a prolonged road to recovery to usual state of health, leading to absenteeism from work and poor quality of life [19]. Significant association was demonstrated between the presence of at least one persistent/new symptom in post-COVID-19 patients to mode of isolation (Table 6), whereby patients who were treated in an intensive care unit had higher chances of getting at least one prolonged symptom beyond two weeks from diagnosis of COVID-19. Post intensive care syndrome is a well-defined morbid entity, which has been described in patients after discharge from an intensive care unit following severe illnesses. However, post-COVID syndrome has been described not only in patients who were discharged from intensive care but also in those that did not seek medical help and recovered at home [18]. Thus, our finding regarding the association of mode of isolation to persistence of post-COVID symptoms needs further evaluation.

The emergence of multiple variants of SARS-CoV-2 in recent times is a matter of grave concern that raises the question of a possible escape from vaccine-induced immunity and could be a major driving force for perpetuality of the current pandemic. Healthcare workers need to be aware of the persistence of multitude of symptoms in COVID-19 patients post discharge, and ways to mitigate it. The physical and psychological burden on survivors of this novel disease is yet to be addressed in a holistic way. Despite the fact that the scientific community is still in search for a breakthrough to save lives during the acute phase of the disease, the post-COVID morbidity of millions cannot be ignored. This study addresses the post-COVID symptomatology in one of the tertiary care centers in Nepal and represents data from low- and middle-income countries in South Asia. With a third wave of COVID-19 looming round the corner in South Asia, more studies need to be planned to investigate the long-term consequences of COVID-19.

This was a single-centered study and had a small sample size. Larger studies need to be conducted to know the actual burden of persistent symptoms in post-COVID-19 patients. An important point of consideration is that this was a hospital-based study; thus, the actual prevalence of post-COVID-19 symptoms might have been overestimated. A single follow-up visit was only considered in this study. Longer duration of follow-up is warranted to study the evolution of persistent symptoms in post-COVID-19 patients.

## 5. Conclusions

This study concluded that there was a high prevalence of persistent symptoms in post-COVID-19 patients. Post-COVID-19 symptoms had no association with age group, gender, presence of any comorbidity, or severity of disease. Healthcare workers need to acknowledge this fact and be aware of the long haulers of COVID-19 so that necessary therapeutic and rehabilitative services can be offered to such groups of patients. Our findings highlight the need for long-term monitoring of COVID-19 patients post convalescence, to understand the implications and consequences of persistent symptoms in the well-being of these apparently recovered COVID-19 patients.

## Figures and Tables

**Figure 1 tropicalmed-06-00113-f001:**
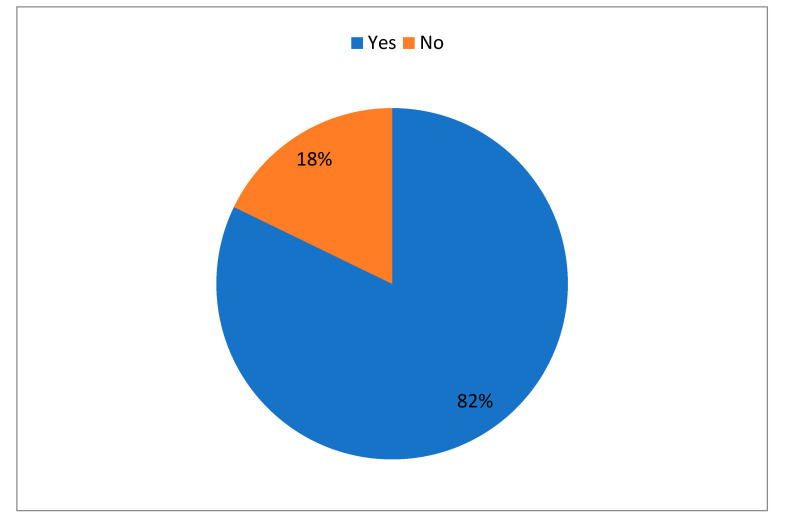
Distribution of post-COVID patients with at least one persistent/new symptom over two weeks after acute COVID-19 infection (N-118).

**Figure 2 tropicalmed-06-00113-f002:**
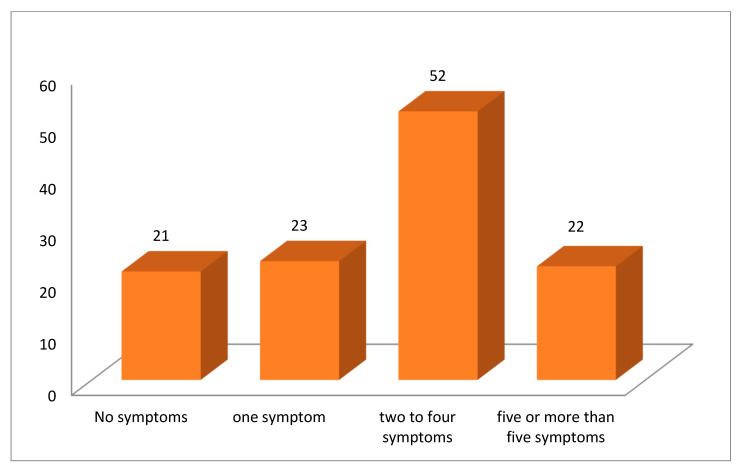
Distribution of post-COVID patients with the number of persistent/new symptoms during their first follow-up visit (N-118).

**Table 1 tropicalmed-06-00113-t001:** Baseline characteristics of post-COVID-19 patients presenting to follow-up OPD at STIDH (N-118).

Baseline Characteristics		Frequency *n* (%)
Gender	Male	81 (68.6)
	Female	37 (31.4)
Age group	less than 30 years	8 (6.8)
	30 to 59 years	76 (64.4)
	60 to 74 years	26 (22)
	75 years and above	8 (6.8)
Comorbidities	Hypertension	29 (24.6)
	Diabetes mellitus	17 (14.4)
	Chronic respiratory diseases	9 (7.6)
	Heart disease	6 (5.1)
	Hypothyroidism	8 (6.8)
	Psychiatric illness	5 (4.2)
	Cerebrovascular disease	1 (0.8)
	Seizure disorder	1 (0.8)
	Rheumatoid arthritis	1 (0.8)
	None	63 (53.4)
Time since diagnosis of COVID-19 at first-follow up	3 to 4 weeks	79 (66.9)
	5 to 6 weeks	25 (21.2)
	7 to 8 weeks	9 (7.6)
	9 to 12 weeks	3 (2.5)
	More than 12 weeks	2 (1.7)

**Table 2 tropicalmed-06-00113-t002:** Symptomatology during COVID-19 illness of post-COVID patients presenting to follow-up OPD at STIDH (N-118).

Symptoms	Frequency *n* (%)
Fever	88 (74.6)
Myalgia/Body ache	88 (74.6)
Cough	87 (73.7)
Shortness of breath	71 (60.2)
Anosmia/Hyposmia	58 (49.2)
Ageusia/Dysgeusia	54 (45.8)
Headache	50 (42.4)
Chest pain	38 (32.2)
Decreased appetite	37 (31.4)
Diarrhoea	34 (28.8)
Sore throat	30 (25.4)
Runny nose	24 (20.3)
Nausea/Vomiting	17 (14.4)
Dizziness	14 (11.9)
Abdominal pain	7 (5.9)
Hemoptysis	3 (2.5)

**Table 3 tropicalmed-06-00113-t003:** Course of COVID-19 illness in post-COVID patients presenting to follow-up OPD at STIDH (N-118).

Course of COVID-19 Illness		Frequency *n* (%)
Severity of COVID-19	Mild	43 (36.4)
	Moderate	15 (12.8)
	Severe	60 (50.8)
Mode of isolation during COVID-19	Home	46 (39)
	Ward admission	34 (28.8)
	ICU admission	38 (32.2)
Duration of hospital admission	None	46 (39)
	less than one week	20 (16.9)
	one to two weeks	43 (36.4)
	three to four weeks	6 (5.1)
	more than four weeks	3 (2.5)
Highest oxygen delivery device used during hospital stay	None	58 (49.2)
	Nasal cannula	44 (37.3)
	Face mask	9 (7.6)
	Non- invasive ventilation	7 (5.9)
	Invasive ventilation	0 (0)
Use of COVID-19 specific medications during hospital stay	Oral or intravenous steroids	65 (55.5)
	Remdesivir	68 (57.6)
	Convalescent plasma therapy	14 (11.9)

**Table 4 tropicalmed-06-00113-t004:** Persistent/new symptoms in post-COVID-19 patients presenting to follow-up OPD at STIDH (N-118).

Symptoms	Frequency *n* (%)
Shortness of breath	48 (40.7)
Fatigue	39 (33.1)
Chest heaviness	33 (28.0)
Cough	32 (27.1)
Chest pain	23 (19.4)
Insomnia	19 (16.1)
Anxiety	16 (13.6)
Anosmia/Hyposmia	11 (9.3)
Palpitations	11 (9.3)
Ageusia/Hypogeusia	8 (6.8)
Decreased appetite	7 (5.9)
Headache	4 (3.3)
Throat discomfort	3 (2.5)
Burning sensation along limbs	3 (2.5)
Impaired concentration	2 (1.7)
Hot flushes	1 (0.8)
Parosmia	1 (0.8)
Burning sensation in the perinostrillar area	1 (0.8)

**Table 5 tropicalmed-06-00113-t005:** Oxygenation status of post-COVID-19 patients presenting to follow-up OPD at STIDH (N-118).

Oxygenation Related Parameters		Frequency *n* (%)
Grade of dyspnea during first follow-up	MMRC 0	70 (59.3)
	MMRC 1	29 (24.6)
	MMRC 2	14 (11.9)
	MMRC 3	5 (4.2)
	MMRC 4	0 (0)
Use of domiciliary oxygen during the first follow-up visit	Yes	16 (13.6)
	No	102 (86.4)
Oxygen saturation at rest during the first follow-up visit	≥94%	94 (79.7)
	90 to 93%	15 (12.7)
	85 to 89%	3 (2.5)
	80 to 84%	4 (3.4)
	75 to 79%	2 (1.7)

**Table 6 tropicalmed-06-00113-t006:** Association of the presence of at least one persistent/new symptom in post-COVID-19 patients with baseline characteristics (N-118).

		Presence of at Least One Persistent/New Symptom in Post-COVID Patients		*p* Value
		Yes	No	
Gender	Male	68	13	0.463
	Female	29	8	
Age groups	Less than 30 years	7	2	0.177
	30 to 59 years	58	17	
	60 to 74 years	23	2	
	More than 75 years	9	0	
At least one comorbidity	Yes	45	10	0.919
	No	52	11	
Severity of COVID-19	Mild	31	12	0.056
	Moderate	12	3	
	Severe	54	6	
Mode of isolation	Home	34	12	0.040 *
	Ward	27	7	
	ICU	36	2	

* *p* value < 0.05.

## Data Availability

The datasets supporting the conclusions have been included in this article. Source data can be made available on request by the corresponding author.

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
