# Peer review of "Persistent Symptoms in Post-COVID-19 Patients Attending Follow-Up OPD at Sukraraj Tropical and Infectious Disease Hospital (STIDH), Kathmandu, Nepal"

_tropicalmed, 2021, doi:10.3390/tropicalmed6030113_

Round 1

Reviewer 1 Report

In this article authors showed the high prevelence of persistent symptoms in post-COVID-19 patients. Health care workers should acknowledge and be aware about long haulers of COVID-19 so that necessary treatment can be provided and also highlighted the need for follow up of such patients following discharge, in order to assess and mitigate long term implications.

Minor comment

I would recommend a further reference in regards to the incidence of post-COVID-19 syndrome in discussion, preferably following the lines 165-167, "Pavli A et al. Post COVID syndrome: Incidence, clinical spectrum, and challenges for primary healthcare professionals. Arch Med Res. 2021

Reviewer 2 Report

Little has been done to address the potential long term health implications for more than 95% COVID-19 patients who have already recovered from this novel disease worldwide.  Some comments have been presented as follows.

  1. Data collection
  • The validity and reliability of the data collection have to be addressed. For example, intra- and inter- rater reliability.
  •  The variables related to persistent or new symptoms in post COVID-19 need to be stated in detail.
  1. Data analysis
  •  The variables are divided into categorical variables and continuous variables. The inferential analysis was done using chi-square and Fisher’s exact test. How can this kind of statistics be applied to the continuous variables?  The management and proceeding of variables need to be described in the text.
  •  The sum of comorbidities, age and duration are continuous variables.  The sum of persistent symptoms is also continuous variables. The associations of variables described above could be investigated.  It does not need to be transformed to the categorical variables.  Continuous variables are more precise than the categorical variables.
  1. Results
  •  The abbreviation of MMRC (p.5) needs to be written at the first time in the text.
  •   The management and proceeding of variables in Table 6 need to be described in the text (p.6).
  1. Discussion

The authors need to explain or deliberate on the variables’ characteristics and proceed the data again.  Then the focus of discussion can be based on the findings.

  1. Conclusions

 The related factors of persistent symptoms in post COVID-19 patients can be added to the text.

Round 2

Reviewer 2 Report

Please check the spelling and grammar again. For example, line 285, ... beyond the acute phase of disease proper.
